# The Human Ecological Perspective and Biopsychosocial Medicine

**DOI:** 10.3390/ijerph16214230

**Published:** 2019-10-31

**Authors:** Felix Tretter, Henriette Löffler-Stastka

**Affiliations:** 1German Society for Human Ecology, A-1040 Wien, Austria; edu@felix-tretter.de; 2Department of Psychoanalysis and Psychotherapy, Medical University Vienna, 1090 Vienna, Austria

**Keywords:** human–environment, relationship, professionalism, competence, mental health, psychic development, change processes, therapists and medical doctors, socialization, lifestyle

## Abstract

With regard to philosophical anthropology, a human ecological framework for the human–environment relationship as an “ecology of the person” is outlined, which focuses on the term “relationship” and aims to be scientifically sound. It also provides theoretical orientations for multiprofessional clinical work. For this purpose, a multi-dimensional basic grid for the characterization of the individual human being is proposed. The necessity and meaningfulness of a differentiation and systematization of the terms “environment”, and above all “relationship”, are demonstrated, and practical examples and links to similar framework models are given.

## 1. Integrative Concepts of Health and Disease by Human Ecological Medicine

More than 40 years ago, George L. Engel [1] proposed his influential view of a three-dimensional bio–psycho–social systems model that helps understand health and disease in the context of “psychosomatics”. In parallel, the issue of “environmental health” came up, emphasizing the health effects of the physico-chemical environment. At present, “theory-free” multifactorial analyses and models dominate epidemiological research and biochemical experiments consolidate current medical knowledge. In order to re-establish a multi-facetted but integrated theoretical view on health and disease, a human–ecological framework is proposed. Human ecology, with a view to philosophical anthropology, is concerned with the study of the human–environment relationship and can be characterized as “the ecology of the person”. It focuses on the central term “relationship” and its variations, and also offers a theoretical orientation to the multiprofessional practice of clinical work as it was practiced in the field of addiction treatment and prevention. A multidimensional conceptual framework was proposed for the characterization of the individual human being as well as the concept “environment”, and above all, “relationship”. Practical examples are included and links to similar framework models are mentioned. 

## 2. Integrative Concepts of Health and Disease 

*Biomedicine* has been a successful research approach in medicine. However, psychological and social factors also influence health and disease. For this reason, more than 40 years ago, George L. Engel proposed his influential view of a three-dimensional *bio–psycho–social systems model* that helps understand health and disease in the context of “psychosomatics” [1]. At that time, the health influences of the physical environment, natural and artificial factors also became interesting issues and the field of environmental health was created, more or less explicitly excluding the psychosocial domain [2,3]. Since then, “theory-free” multifactorial analyses and models dominate the *epidemiological* and *clinical research*, mixing variables with different ontologies without reference to a conceptual framework that deserves the name “theoretical medicine”. Concepts such as “vulnerability”, “resilience”, “risk factors”, “salutogenesis”, “protective factors”, etc., characterize these approaches and the construct-related theoretical discussions that are conducted in order to “explain” the occurrence of diseases. Even big data will not succeed in better explanations of diseases if they are not embedded in “big theories” [4].

At present, the health sciences are aware of the health effects of climate change and other environmental issues and are seen in a systemic framework capturing interactions, feedback loops, etc. These frameworks represent conceptually socio-ecological systems and integrate physical and social issues, and therefore the term “ecology” is sometimes also used [5]. Some issues, like urban health touch Public Health [6], other are very urgent such as climate change and water supply, food security, and population health and are connected with the main goals of the UN Program of Sustainable Development [7,8]. As a consequence, there is again some evidence for the need of an extended integrative medical model [9,10,11,12]. Some researchers and institutions have also used the academic label “(social) ecology” [13]. This later view will be the focus of our paper, which proposes a *human ecological conceptual framework* that covers several perspectives touched on above such as the individual person as well as the population level, the natural, social, and built environment and the household of human–environment interactions [14].

## 3. Preliminary Philosophical Remarks 

Integrative modeling requires interdisciplinary integrative *epistemology* and bridge concepts. This has been discussed lately regarding the brain–mind gap in neurosciences [15]. Especially in the context of health sciences, the integration of the “objective” perspectives of research and “subjective” perspectives of patients have to be connected. Regarding this issue, *constructivism* offers a common view point in the social and human sciences: it is supposed that there is no strategy to determine “real reality” except through discourses. Plato has already pointed this out with his cave parable. More recently, Gregory Bateson, Paul Watzlawik, Heinz von Foerster, and Ernst von Glaserfeld have worked out this position in detail [16]. Nevertheless, constructivism admits that “out there”, there is something (landscape), referring to which the construction of the outside world (the map) is “viable”, (i.e., has to be corrected by obstacles if necessary, but otherwise makes actions possible). To exchange these maps intersubjectively, language is essential and “reality” is co-constructed in discourses of the stakeholders of the respective problem [17,18]. In the practice of health services, experts co-construct the concepts of health or disease with the client or patient. However, in this relationship, the experts (e.g., medical staff) in most cases have better justified—scientifically based—access to the truth. However, regarding drug experiences, the experts do not comprehend fully the subjective experience of the user. This indicates that “scientific realism” (or “constructive realism”) might be a more appropriate epistemological position, an issue that needs further discussion [19,20]. Subsequently, scientific theoretical aspects such as the difference between description, explanation, and prognosis must be taken into account [21]. 

Closely connected with these epistemological issues is the current, sparsely developed field of the concept of different organizational levels of the world as it was discussed as *ontology*, which was probably last elaborated by Nicolai Hartmann [22]. Remarkably, today the Big Data approach enters into this conceptual gap by attempting to associatively cover all areas of the individual “life world” of as many people as possible and to “deduct” behavior clarifying contexts in the form of hierarchical concept structures that are designed for pragmatical and theoretical reasons [23,24]. In this conceptual model, we tried to explicate several important dimensions of consideration such as the “human”, the “environment”, and the “relations”, topics that can be described by common basic properties but need diverse conceptual differentiations. For this reason, the various lists for the consideration of these subjects differ regarding their conceptual resolution.

Reference is made here to a further, very central area of philosophy, namely *philosophical anthropology* as general anthropology, insofar as the basic characterizations of man are addressed regarding his multidimensionality as well as his essential behavioral features. For instance, Max Scheler, Helmuth Plessner, and Martin Heidegger created differentiated foundations of anthropology as a philosophical discipline relevant for the present [25,26]. Finally, the area of *ethics* is relevant, insofar as the question to be clarified, as to how to deal with such an integral recording of people in view of the problem of the electronic recording of everything and everyone, which has increased due in particular to digitalization, with possibilities of the abuse of such information so that patient sovereignty is secured [27].

However, the philosophical breadth, and above all, the necessary depth, especially with regard to the conceptual problems, would have to be worked out somewhere and somewhen separately. 

With these issues in mind, we have to ask, how can the ecological perspective be integrated with environmental and public health issues? Up until now, only a few approaches have integrated public health and environmental health perspectives [28] or applied an explicit social ecological view [29]. Here, we explore a systematized human ecological view point that integrates *somatic* and *psychosocial medicine* as well as *environmental health* and *public health* as a population perspective. 

## 4. What Is Man? – Perspectives of Philosophy

The bio–psycho–social model has one focus on “human” as a socially situated psychophysical being. The focus here is on the human’s relationship to their environment, with regard to their disorders and diseases (i.e., an “ecology of the sick person”, for example, with regard to addictive disorders. This is based on a *multi-level model* that conceptually takes into account the external differentiations of the environment and internal differentiations of human beings (Figure 1) [30,31,32].

From the point of view of philosophy, which can be understood as a kind of metatheory of the sciences, the question “What is man?” can be subdivided both epistemologically (“How can we know what man is?”), and ontologically (”What constitutes man?”), and last, but not least (“What makes man a man?”) as a genuinely anthropological question. In general, one differentiates the question of “the” human being into a diversified pluralization: men and women, children and old people, etc., are differentiated and the question leads to the individualization of the human being. However, regarding the work of Aristotle, in all cases, a three-dimensionality can be identified by emphasizing that the human being is an animal, but a spiritual, reflective animal (Zoon logon echon), and above all, a social animal (Zoon politicon): Man, or rather every person, is a bio–psycho–social being. Ultimately, many philosophers agree that humans must be pragmatically characterized as *multidimensional beings* [33]. This also corresponds to the experiences in clinical practice where each person can be described quite individually, if one considers about *eight relatively autonomous, but causally linked dimensions* of their existence, which are already collected in a qualified anamnesis in the practice of helping professions such as doctors, psychologists, social pedagogues, etc. The central epistemological difference of “objective” and “subjective” data is basically assumed here, and is not mentioned here explicitly [32]: 

(1) *Temporality:* The time of birth, the season, the epoch in which a person is born and spends his childhood (e.g., wartime) shapes the future of life. Temporality, with the present as the interface between past and future, is increasingly experienced in individual ontogenesis as a subjective dimension of the finiteness of existence, and thus shapes experience and behavior. Ultimately, the knowledge of finiteness of human life is a specifically human knowledge that seemingly cannot be experienced in its existential dimension by learning machine algorithms either. However, with their fashions (e.g., fashion drugs), time also shapes concrete existence. Temporality is pathologically relevant regarding the time structure of exposure to adverse agents like toxic environmental chemicals. 

(2) *The place (or space):* Not only the time, but also the place of birth and its geophysical characteristics (climate, landscape) determine the physical living conditions of the person who was born here and not there. Physical places also bring with them certain personal and socio-cultural characteristics that affect the person and are more or less determinant. The *real local reference*, based on these characteristics (e.g., as home life), develops positively or negatively in sum, and in this way, it also determines the disposition for migration (*push factors*) if necessary. The *fictional reference to another place*, fantasies about it, as hopes for instance are, represent *pull factors* that additionally determine the inner disposition to migrate. Spatial properties of exposure to toxic agents—proximity or distality—influence health and disease.

(3) *Physicality (Personal self I):* Genetic and epigenetic individuality as well as physical sex properties and other somatic characteristics, both observable and experienced (“corporeality”; [34,35]), imply options as well as frictions to live. They also condition, but do not determine, health and illness, for example, in the form of physiological stress vulnerability, which can also be predisposed to addiction. 

(4) *The mental (Personal self II):* Already the experienced physicality, the corporeality shapes the behavior and the relationship to the environment. Cognitive competencies and emotional dispositions shape the characteristics of the individual personality, which is also shaped by socio-cultural factors. The guiding structure of behavior is sustainably psycho-socially produced and constructed affective-cognitive schemata. In this way, the basis of socio–psycho–biographical individuality emerges. It should be mentioned here, that it cannot be explained by the brain structure alone [36,37,38].

(5) *Language ability*: This covers the ability to symbolize, to receive and produce, communicate, and the language that is practiced in the social environment. This linguistic competence and the person’s performance in the end shape his or her spirituality and thus also his or her social options and limitations. Already at this point, the important construct of the “structural coupling” between consciousness and the social world should be used, in order to construct integrative conceptual frameworks [39].

(6) *Sociality*: The existential relevance of the external counterpart, the “interpersonal” as a fundamental relation to other people, which first of all concerns those persons who make up the *family* into which the person was born and also the *relatives* surrounding them, they determine from birth the social micro-world as options and frictions of childhood. Each of these people in a single human’s environment contributes to the multidimensional potential of the development of the individual child: empathy for the child or conflicts between parents and the child are known to have a strong influence on the psychological development of the person (e.g., binding problems). Later, *peers* are the relevant reference persons, colleagues, own family, etc. This level of the *microsocial* is superimposed on the *meso-level* of the municipality and the *macro-level* of the institutions of society. 

(7) *Culture*: The embedding of the person in the local, regional, and national system of values, beliefs, knowledge, meaning, morals, ethics, etc.; simply saying “culture” as the content of the social, determines the basic patterns of behavior and feelings as superordinate structures. The cultural elements can be in conflict with each other, but also to the inner drives (knowledge versus faith). They can even be pathogenetically relevant, as Freud emphasized in his *structural model* of neuroses based on the sexual pathology of the 19th century. The cultural immanence not only limits the competence of each expert in practical therapy, but also in research logic, which can only be estimated in an intercultural comparison [40]. 

(8) *Economic basis:* The economic level of the individual household is the basis of living comfort from birth and continues in the situation of being a child with poorer or richer parents, which has a strong influence on the options for education and achieving economic advancement (vertical mobility). However, a very good economy of the family of origin can also influence the opposing refusal of socio-economic wellbeing. It is known that social class-specific inverse correlations of disease risks (e.g., tobacco consumption) are known, whereby educational factors are also taken into account. A philosophical deepening of this dimension seems enriching [41].

Of all these eight dimensional categories, however, only the physical, the mental, and the internalized social of the social world are directly related to the person, which is also reflected partially in the already mentioned bio–psycho–social model’ of medicine [1]. Time, place, personal environment as a component of the external social, culture, and economy are external dimensions (or factors) that can be attributed to the “environment” into which man is “thrown” in the sense of Heidegger. Gathering information about these dimensions must always distinguish self-reports and “objective” data and relate them to each other.

## 5. What Is the “Environment”? 

The differentiating emphasis on the human being implies the concept of the “environment” as a surrounding exterior: If we talk about men, the question raises “where” and so we have to already consider the surrounding space. In *everyday language*, however, this expression is often referred to the *natural environment* and in the *social scientific context* to the *social environment*. The terms “surroundings”, “setting”, etc. are also often used. Highlighting environment as a process, “factors”, “dimensions”, “areas”, “levels”, etc. of the environment are differentiated. However, there is no generally accepted convention on the taxonomies of these concepts. Additionally, there are no binding interdisciplinary language regulations. The conceptual heterogeneity could only be resolved by consensus conferences. Nevertheless, it is helpful, especially with regard to the ecological perspective ultimately presented, to consider the following conceptual distinctions and to use them as a framework [31]: 

(1) *Subjective versus objective environment*: Depending on the epistemological position of the observer, the environment can be determined by physico-chemical variables and methods or from a subjective phenomenological perspective as the experienced environment (e.g., measured and/or experienced air temperature). This difference is also reflected in the epistemic difference between the view of academic ecology, as defined by Ernst Haeckel in 1866 (“surrounding outside world”), and environmental theory, as constructed by Jakob von Uexküll: Environment (Umwelt) is an entailment of the perceptual and operational world (*Merkwelt* and *Wirkwelt)* of the individual [42,43]. In more recent times, other subject-centered concepts such as “living space”, as some kind of an internal representation according to Lewin [44] and the ”life world”, according to Schütz and Luckmann [45,46], have become theoretically and practically significant for social scientists. 

(2) The *temporal dimension* can be seen in the heuristic usability of the concept of “former”, “current”, or “future environment”. 

(3) The next intuitively understandable basic dimension is the *space or place*, which can be differentiated according to “ranges” such as *macro-environment*, *meso-environment*, and *micro-environment*, in each case “objectively” or subjectively conceptualized. In the context of human ecology, for example, in the sense of Uri Bronfenbrenner, the microenvironment (or micro-system) is primarily understood as the family, but also school or work, the leisure area, and other sub-areas of the individual’s living space, while the meso-environment (or meso-system) is the summary of these microenvironments. For its part, it is again embedded as a middle level in the macro-environment that is called the macro-system [47].

(4) *Entities:* material-energetic or physical versus immaterial (informational) environment and other sub-domains can be distinguished such as an *inanimate natural (abiotic), animated natural (biotic), technical*, *personal*, *social*, *cultural*, etc., environment. The everyday understanding of these terms is sufficient here, but a precise definition remains difficult, especially since there is currently no *philosophical ontology* (see above). 

(5) *Areas of life*: Living environment, working environment, etc., are environmental areas that are differentiated according to their *functional significance for the person*. They are dissociated, but partially overlapping fields of life. 

(6) *Qualities:* They can be classified as “good” or “bad” in terms of the environmental effects experienced by the person. 

(7) *Quantities*: These properties of the environment can be represented, for example, as the *density* of the respective environmental elements occurring in the space–time framework. 

(8) *Effects and their directionality*: impacts, effects, and interactions are corresponding categories. 

This conceptual apparatus for the epistemic object “environment”, which, as mentioned, usually occurs *implicitly* in the most diverse texts, shows references, for example, to Pierre Bourdieu’s concepts such as “social space” or “fields”, an aspect that needs to be clarified more precisely, especially since Bordieu acknowledges an objective social reality and not only its constructedness by the subject as constructivists often see it. Thus, from a pragmatic point of view, the characterization of any individual person by their environmental relationship or life situation is easy to show.

## 6. What Are “Relationships”? 

The categorical distinction between human and the environment also implies the term “relationship”: the separate must be related to each other again in order to ensure proximity to reality. Similar to the concept of the environment, a fundamental *double perspective* of the human–environment-relationship must be considered, namely that from the point of view of the external observer (third person perspective) and also that from the inner point of view of the experiencing subject (first person perspective).

Colloquially, the term “relationship” is usually used to describe the relationship between people, (i.e., contact), the (interpersonal) relationship, etc. This expression primarily denotes the experienced closeness to another person (i.e., a spatial or emotional relationship), or also its “meaning” (i.e., the effect on one’s own life). In this view, the shape of the person’s motor behavior toward the environment, as a pattern of lifestyle, is also a form of person–environment-relationship: the person’s motor behavior such as walking or staying, speaking or being silent, or taking, giving, or refusing something, etc., manifests the person’s (inner) relationship to his physical and/or social environment. In this view, the *lifestyle*, related to a certain life domain, is a relationship pattern of acceptance, appropriation, approach, distance, acceptance, rejection, etc. For example, in this abstract understanding, *mobility styles* are patterns of *space reference* and *time reference*, in particular being accelerated on the road, resting in yourself, etc. It is a personal environmental relationship regulated by the person; however, the perception or behavior, the impact of the environment and its effect on behavior can also be formulated quite precisely in categories of the concept of relationship. 

The term relationship, as presented here, is thus one of the most abstract concepts of all and therefore is well suited for a more comprehensive taxonomy based on the mentioned dimensions of being human: 

(1) The *epistemology* must be explicated fundamentally and a distinction made between “subjective” (experienced) relation and “objective” (observed) relation must be made. 

(2) *Temporality*: Current relationships, past relationships, expected future relationships, etc., should be defined. 

(3) The *locality* (or spatiality): Here it is, above all, the proximity and distance of the person to the environmental elements, also metaphorically or topologically: local relations, far-reaching relations, focal relations, internal relations, external relations, proximity relations, etc. An elementary spatial reference is the experience of (sweet) “home”. In addition, the relationship of the person to the subjects in social space and their demarcation creates identity, for example, during puberty, and is constituted by the person’s experiences of interaction. This aspect is modeled by the psychoanalytical *object relationship theory* as the *inner representations of the self and the objects of the environment* (especially the primary caregiver) and their relationships to each other, mainly regarding proximity and distance [48]. For psychiatric practice, the more precise spatial analysis of the social life of patients in the 1980s was already discussed under the heading “ecological psychiatry” [49,50].

(4) *The modality*: Physical-material, physical, mental, etc., can be used here to differentiate modes of relations. 

(5) *Directionality*: Relationships “away from” or “toward”, self-referral relationships, etc. should be distinguished. Simplified, one can also speak of “giving” at action with an “away from” and of “taking” when actions are directed “toward” something, especially when it comes to interactions with objects. 

(6) *Quality*: A distinction can be made between “positive” or “good”, “negative” or “bad” relationships, etc., according to the effect. The topic of quality has to be discussed under the aspects of psychosocial development, for instance, the quality of relationships and its development can be investigated via micro-affective processes in terms of how they occur in the interactions (e.g., between caregivers and their children), as an important component of the relationship structure or modality. The development of modalities depends in what ways the interactional behavior of the caregiver–child pairs differs. The development of the extent to which differences in the parental mentalization ability and the mental health of the children occur, and then are related to their relationship modality/behavior, the quality of relationships matures. 

(7) *Intensity*: Strong and weak relationships, etc. are useful differentiators. 

(8) *Quantity*: Attributes would be “many” or “few” relationships.

(9) *Frequency*: This aspect may be characterized by the term “frequent” or “rare”, etc. 

(10) *Effect*: Relationships can have increasing or inhibiting effects on a system state.

(11) *Function*: A distinction can be made between, for example, “stabilizing” or “destabilizing” relations regarding an organismic function. 

(12) *Contexts*: Depending on the person’s relation to the environment, work, family, etc. can be differentiated according to the person’s domains of life. 

## 7. The Human Ecological Perspective in the Bio–Psycho–Social Helper Practice 

On the basis of this conceptual apparatus of “person”, “environment”, and “relationship”, a human ecological perspective can be developed systematically. Only a few approaches have integrated public health and environmental health perspectives [28] or applied an explicit social ecological view [29]. “Ecology” is drawn in the shortest, but most accurate form as the science of the “household of nature”. Since its foundation by Ernst Haeckel in 1866, however, as a university discipline it has experienced a great differentiation as well as wide adoption in other disciplines [42]. Particularly in *sociology* as a human science, the population ecology approach of *biology* (“Who eats whom”?) was adopted into urban sociology as early as the 1920s in the form of “human ecology” (“Human Ecology” or “Social Ecology”) [51,52]. Relatively independent of this, in *psychology*, Kurt Lewin, in particular, conceived the ecological perspective on the basis of his field, psychology [44], which was further expanded by Uri Bronfenbrenner [47]. In the field of social pedagogy, an individual-centered ecological perspective several times was also articulated [53,54,55,56].

Here, it was assumed at first that the term “household” characterizes best what this integrative meta-perspective means with a relationship structure within the framework of an “Ecology of the Person” [31]. 

### 7.1. "Household" as Relationship Relations and Stress as Disturbance of the Person’s Relationship Household 

The term “household” is thus understood here with regard to health and illness as the “relationship of relationships” or of “interactions” (i.e., as the relationship of relationships between person and environment). If the relationship is also conceptualized as directed from the environment, then there is a *relationship of giving and taking*, with different forms that extends over several levels. In the interpersonal field, *asymmetries* often arise as *pathogenic socio-emotional imbalances*: I give more than I get, I can’t accept offers, etc.; I feel overwhelmed by the environment, whose expectations I only suspect or which are explicitly formulated to me, etc. Offers of the environment concern spatial (housing), personnel (assistance), social (entitlements), cultural (entertainment), and economic (financing). It is at these levels that the demands of the environment (attention, attention, space, money, time, etc.) take place. Additionally, the offers of the person or their demands on the environment are of concern among other things at such levels. 

Therefore, it is about the *question of balance* or, according to Klaus Grawe, about “congruence” [57]: a person continuously experiences that he gives more than he gets, or that more is taken from him than he gets. On the basis of these imbalances of interpersonal relationships, disturbances of the person with subsequent negative (e.g., aggressive) interactions arise. If one starts out from a (dynamic) *equilibrium concept* with regard to the relations of relationships, then (chronic) stress can also be described differently in this way, namely as the product of *persistent relationship imbalances* (see below). Many examples can be found in everyday interaction, whose form shapes the state of mind of the actors, based on dynamizing imbalances (incompatibilities, dissent) and stabilizing equilibria (compatibilities or consensus) (Table 1). 

This situation can be illustrated even more clearly in the form of *visualizations*: If one already considers the *directionality of arrows (and other directed signs) in the relevant diagrams*, then the translation into the words “give” or “bid” can be made with regard to the source of the arrows (Figure 2). As far as the aim of the arrow is concerned, the term “take” follows (e.g., accept). The latter in particular, however, must usually be explained, because in the case of person–environment-relationships in the social sphere, there are situations where offers are rejected and as a result, a disturbance and annoyance, sadness or withdrawal of the other can be triggered in the psychological domain of the person. These emotional distractions can turn into psychological disorders requiring treatment and vice versa: The *therapy* can *interpret emotional problems* such as *anger*, *fear*, and *sadness as an environmental relationship problem* of the person and work on them in a differentiated way. 

### 7.2. Life Area Structure Model

The person’s state of health depends to a large extent on their relationship to the most important domains or areas of life such as their home, family, work, and leisure time (Figure 3). Sommerfeld calls the entire structure, which consists of individual action systems and is related to the individual areas of life, the “life management system” [58]. It is about the integral of what has been experienced in these areas, cumulated over a certain period of time and it is the way it makes you feel about life. 

### 7.3. Lifestyle Patterns of Regulating Relations to Life Domains

Every day, typical patterns of action as personal forms of relationship to the environment are called “lifestyles”, especially in the context of the social sciences [59]. Additionally, in medicine, the pathogenic and salutogenic importance of lifestyles in their weight when compared to genes and to the social situation has been recognized, even when it is considered that the *nutrition style* is a component of the *overall lifestyle*, and together with the “movement style” can have significant effects on body weight. 

This means that it makes sense to relate the construct *lifestyle* to several *domains of life* (Figure 4): Housing style, family patterns, nutrition style, work style, etc. For example, a health-related recommendation to change eating habits will also have to affect patterns of locomotion to be effective. 

### 7.4. An Ecological-Cybernetic Lifestyle Model 

The following conceptual components make it possible to constitute a control loop model of lifestyle with actual values and target values in order to understand the *dynamic dimension* in the treatment context (Figure 5): 

(1) Life situation/life areas (actual values)

As above-mentioned, the areas of life include work, leisure, and family. Their interrelationship is expressed among other things in the work–life balance (or leisure/work balance) disturbed in context of the *burnout syndrome*. 

(2) Life objectives/life plan (target values)

This virtual or visionary level is causally based on wishes and hopes that come about endogenously or exogenously. 

(3) Attitude to life (result)

The reference of the actual to the target results in the life feeling, which can manifest itself in stressful tension in the case of persistent and/or large discrepancies. 

(4) Lifestyles as patterns of living (behavior)

The lifestyles serve the regulation of the life situation with regard to the life feeling and/or the life plan lying behind it. Inasmuch as the life plan is ambitious, the program of the lifestyle is to be as fast and strong as possible: Coffee-to-go and every other “walking structure” is aspired to and makes it selectively considered efficient. 

(5) “Food”/instrumentalize, “Techniques”

At this point, drugs are seen as instruments of lifestyle because they modulate the attitude toward life or seem to help to realize other goal-compatible lifestyles. 

Taken as a whole, this model provides a dynamic view of the ecology of the person and the pathology (Figure 4). Therapeutically, this control loop can therefore be processed at any stage of the process. 

## 8. Conclusions

A human–ecological version of an *integrative framework model* for the *understanding of health and disease* offers a perspective across the individual sciences on the basis of an *interaction model of person and environment*. It focuses on the individual person (“man”), but it can be transferred easily to the population level (“men”). Additionally, different model variants are possible as it was demonstrated. They serve, on the one hand, to classify the variety of conditions of disease, and on the other hand, to provide a multidimensional and inter-professional orientation for diagnosis, planning, organizing, and managing the recovery of the clients. These models make it possible to look from the person concerned to the environment and, above all, to the *relationship structure*, because for a health-promoting situation, the limits of the individual adaptation of the *person–s lifestyle* must be supplemented with *changed environmental conditions*. With this perspective, *human ecology* (and/or social ecology) have already gained wider acceptance in the health sciences in the area of public health and health promotion, especially in the USA [60,61].

Although the human ecological perspective might offer an integrative view, one has to consider the consequences of the current *digitalization* of everything and everyone, which, with such integral descriptive grids, by representation of individual persons perhaps a final step into “technological totalitarianism” is done. Selections in professional life, in particular, can obtain new support in the area of grey data management, because health data are the most valuable commodity on the data market in terms of price, every company manager would like to know whether they have a “risky” person in their staff.

## Figures and Tables

**Figure 1 ijerph-16-04230-f001:**
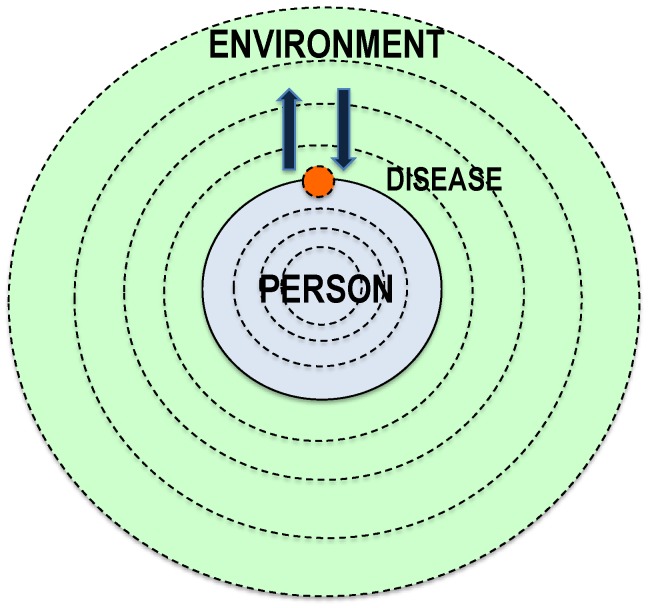
Guiding model of the inner and outer complexity of the human-in-the-world (integrative onion-shell model), which understands diseases and/or disorders (e.g., addiction) as a consequence of inconsistencies in the external and/or internal relationship structure of this system [32].

**Figure 2 ijerph-16-04230-f002:**
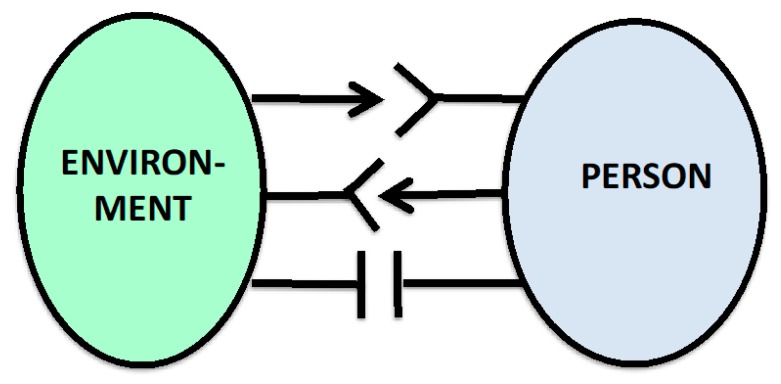
Person–environment relations as a structure of single or double give–take relations, or give–give, or take–take relations, and additional reciprocal rejection relations. The overall relationship can result in pathogenic imbalances.

**Figure 3 ijerph-16-04230-f003:**
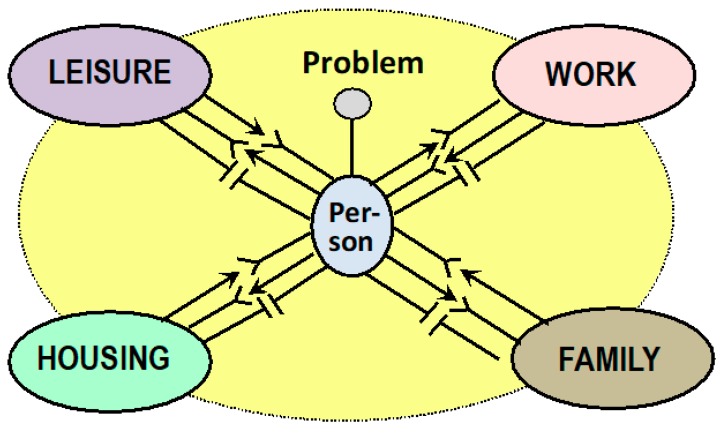
Conceptual framework for a systemic understanding of the reciprocal relationship of the person to their environment with areas of life as the *life world*. For every micro-area—housing, family, work, leisure—there are give–take relationships, which in their totality can co-determine the psychophysical state of health and thus the health of the person or causes of illness. Health is therefore finally the product of comprehensively successful person–environment fits.

**Figure 4 ijerph-16-04230-f004:**
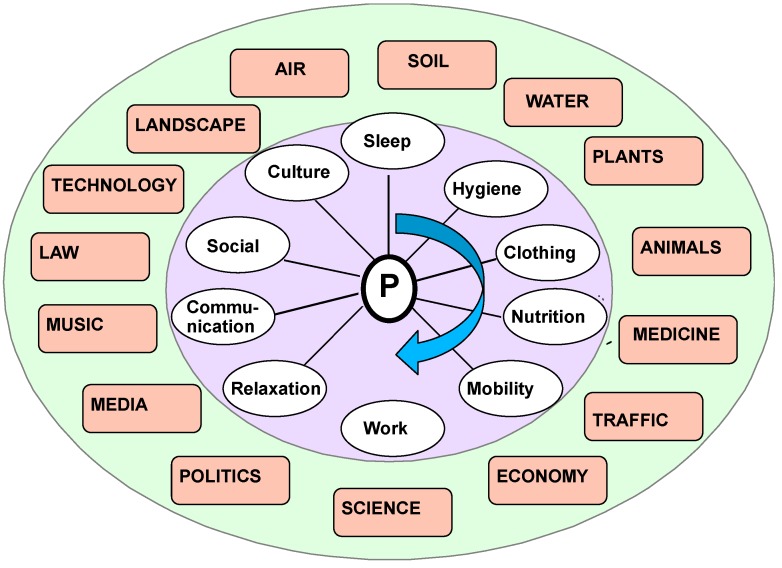
Lifestyle as a system of human–environment relations on a micro-level referred to a daily lifecycle and embedded in macro-level life conditions: From morning hygiene to dressing, nutrition, mobility, and work life over leisure time based social relations communication, and real social life to final cultural issues and health related issues of domain-specific lifestyles are to be considered. This framework helps therapists to design a multi-professional lifestyle related treatment plan for the person.

**Figure 5 ijerph-16-04230-f005:**
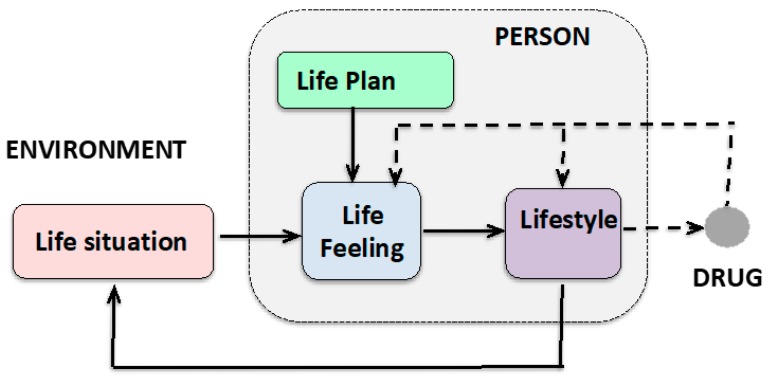
Closed-loop model of the lifestyle as the resultant of the attitude to life, which is based on the relationship between life situation and life plan.

**Table 1 ijerph-16-04230-t001:** Everyday language terms for interpersonal exchange processes that can be understood abstractly as directed relationships that constitute the desired (dynamic) equilibrium in the fundamental *give–take dialectic of the social*.

Give	Content	Take
give out, spend	acknowledgement	accept
give away, devote to	rejection	acquiesce
give off	burden	accept
offer	care	demand
give	money	accept
give away	Furniture	accept

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
