# Peer review of "The Human Ecological Perspective and Biopsychosocial Medicine"

_ijerph, 2019, doi:10.3390/ijerph16214230_

Round 1

Reviewer 1 Report

The paper contains a bold attempt to integrate  widespread and different topics. due to this, it must be seen as a starting point for further works and cannot be handled as a "solution" to a single "question". As this I think itoffers a lot of interesting proposals even so it suffers inevitably from contingency. On the other hand it cannot offer more than  cursory remarks so it is not easy to evaluate what can come up from this attempt of integration. Perhaps it would be helpful to go deeper into some examples (as tried with the problem of drug addiction, 403ff).

Three structural remarks:

There is a certain lack of integration of the different threads. For example: I don't see what follows of the claimed constructivist base of the model in the course of the discussion. Some of the acclaimed positions are extremely shortcuts (what are the achievements and risks of constructivism?). Some of the explanations are lists of aspects which do not operate on the same level  (119ff) and some of these lists are to long (262ff).

Author Response

Open Review

English language and style

( ) Extensive editing of English language and style required 
( ) Moderate English changes required 
( ) English language and style are fine/minor spell check required 
(x) I don't feel qualified to judge about the English language and style 

Yes

Can be improved

Must be improved

Not applicable

Does the introduction provide sufficient background and include all relevant references?

(x)

( )

( )

( )

Is the research design appropriate?

(x)

( )

( )

( )

Are the methods adequately described?

( )

(x)

( )

( )

Are the results clearly presented?

( )

(x)

( )

( )

Are the conclusions supported by the results?

( )

( )

( )

( )

Comments and Suggestions for Authors

The paper contains a bold attempt to integrate  widespread and different topics. due to this, it must be seen as a starting point for further works and cannot be handled as a "solution" to a single "question".

YES AND NO:

YES, WE TRY TO STIMULATE DISCUSSION OF INTEGRATIVE CONCEPTS IN ISSUES OF HEALTH AND DISEASE (PH, EH ETC.). THE CONCEPT PRESENTED HERE CLAIMS TO OFFER MORE OPTIONS FOR INTGRATION.

NO, AS PROGRESS WAS MADE IN INTEGRATING THE THINKING AND ACTING IN CONTEXT OF MULTIPROFESSIONAL WORK IN ADDICTION BUT IT WAS NOT PUBLISHED IN ENGLISCH. THE KEY ISSUE IS TO UNDERSTAND HEALTH AND DISEASE AS A RESULT OF A RELATIONAL DISORDER OF THE HUMAN-ENVIRONMENT RELATIONSSHIP   (E.G. TRETTER “ÖKOLOGIE DER SUCHT” 1998). IT ALSO SERVED AS A FRAMEWORK IN ENVFIRONEMENTAL HEALTH AS “ECOLOGICAL MEDICINE” - IT WAS PROPOSED BY THE AUTHOR IN THE GERMAN DISCUSSION OF ENVIRONMENTAL HEALTH SINCE 1986 AFTER THE TSCHERNOBYL ACCIDENT (PUBLICATIONS IN DEUTSCHES ÄRZTEBLATT).

As this I think itoffers a lot of interesting proposals even so it suffers inevitably from contingency.

=> WE TRIED TO IMPROVE CONTINGENCY BUT THE PAPER SHOULD NOT BE TOO LONG.

On the other hand it cannot offer more than  cursory remarks so it is not easy to evaluate what can come up from this attempt of integration.

Perhaps it would be helpful to go deeper into some examples (as tried with the problem of drug addiction, 403ff).

SEE ABOVE: PRACTICAL INTEGRATIVE THEORY IN ADDICTON AND

=> WE INTEGRATE MORE EXAMPLES

Three structural remarks:

There is a certain lack of integration of the different threads. For example: I don't see what follows of the claimed constructivist base of the model in the course of the discussion. Some of the acclaimed positions are extremely shortcuts (what are the achievements and risks of constructivism?).

WE MADE THIS POINT CLEARER IN THE TEXT: CONSTRUCTIVISM IS AN INTERDISICIPLINARELY WIDLY ACCEPTED EPISTEMOLOGICAL POSITION BUT IT LACKS IN DISTINCTION OF SCIENTIFIC “CONSTRUCTION” AND INDIVIDUAL LIFEWORLD-RELATED CONSTRUCTION - WE THINK THAT THERE IS AN IMPORTANT “DIFFERENCE THAT MAKES A DIFFERENCE” WITH THE CONSEQUENCE THAT A BIMODAL EPISTEMOLOGICAL APROACH IN HEALTH ISSUES (E.G. TO BE SICK OBEJCTIVELY AND FEEL SICK SUBJECTIVELY) IS UNAVOIDABLE. OF COURSE THIS CAN BE CRITICIZED BUT WE NEED A PRAGMATIC DECISION.

Some of the explanations are lists of aspects which do not operate on the same level  (119ff) and some of these lists are to long (262ff).

SORRY, WE COULD NOT IDENTIFY THE REFERENCE NUMBERS IN THE TEXT. HOWEVER, TAKING OVER THESE ARGUMENTS WE EXPLAINED AND JUSTIFIED IT IN THE “ONTLOGY” SECTION OF THE PHILOSPHY CHAPTER.

WE THINK THAT THESE LISTS POSSIBLY COULD BE MADE MORE CONTIGENT BUT THEY ARE BASED ON CLINICAL EXPERIENCE THAT  IS NOT YET RELATED TO “ONTOLOGY OF RELATIONS” THAT PROBABLY DOES NOT EXIST YET.

Reviewer 2 Report

This is a very comprehensive and complex article describing the intricate mechanisms for understanding relationships across human ecological  models. One area that is of concern is the language of gender. For example, at the introduction of the paper, lines 21-34 the term men-environment is used to define ecology of the person.

The term " men" and "man" stays consistent up to page 3, but on page 6 lines 239-244 person environment is used. Is it at all possible to recognize the need for gender inclusive language and stay consistent with person environment relationship?

FIG 4 on page 9 again addresses men -environment and not person-environment and language in conclusion includes man and men again.

I appreciate the definition of What is Man? on page 3, I wonder if there is a way to provide gender inclusive language and pronouns?

Author Response

This is a very comprehensive and complex article describing the intricate mechanisms for understanding relationships across human ecological  models. One area that is of concern is the language of gender. For example, at the introduction of the paper, lines 21-34 the term men-environment is used to define ecology of the person.

--> Thank you very much for the suggestions, we changed to human-Environment consistently in the manuscript.

The term " men" and "man" stays consistent up to page 3, but on page 6 lines 239-244 person environment is used. Is it at all possible to recognize the need for gender inclusive language and stay consistent with person environment relationship?

--> We changed to human-environment, only if the clinical encounter was considered we stayed with person.  

FIG 4 on page 9 again addresses men -environment and not person-environment and language in conclusion includes man and men again.

--> We changed accordingly to be consistent in the whole manuycript.

I appreciate the definition of What is Man? on page 3, I wonder if there is a way to provide gender inclusive language and pronouns?

--> Thank you again for the hint, we changed to human-environment-....